# Unpaired-to-paired data synthesis: Learning to model disease effects via contrastive analysis of neuroimaging-derived features

## Abstract

Advances in machine learning have enabled the analysis of complex, high-dimensional datasets, yet neuroimaging lags behind due to data privacy and sharing constraints. Synthetic data offers a promising solution for developing and training models. However, synthesizing disease-specific datasets is challenging, as neurological disorders induce progressive changes in the brain that are subtle and often obscured by normal brain variability. Contrastive analysis provides a framework to learn generative factors that deconvolve variation shared between background (e.g., healthy) and target (e.g., diseased) datasets from variation unique to the target, making it particularly effective in capturing as well as modeling subtle disease effects. In this paper, we reformulate this framework for the synthesis of neuroimaging-derived features, specifically brain regional volumes from T1-weighted structural MRI. Given unpaired neuroimaging samples of healthy and diseased participants, we learn to generate paired healthy and disease feature representations that emulate real disease effects. We show that paired synthesis enables fine-grained, individual-level modeling of disease effects, improving downstream analyses, and supporting more precise exploration of disease heterogeneity. We validate the models on both semi-synthetic and real-world brain regional volume datasets, specifically designed to highlight the heterogeneity parsing capability of contrastive analysis. The models are available at: [link].

## 1 Introduction

Machine learning and AI systems have facilitated analysis of high-dimensional, complex, multimodal data enabling rapid development in fields of computer vision and natural language processing. Tools that automate workflows and optimize time and labor have transformed many industries, streamlining processes while advancing productivity (Rashid & Kausik, 2024). Beyond efficiency, these systems also provide deep insights into underlying structures and patterns hidden within high-dimensional data, enabling data-driven decision-making and discovery. While areas such as robotics, conversational AI, and autonomous driving have seen widespread applications of these technologies, progress in healthcare has been more constrained.

Healthcare data, due to the sensitive personal information it contains, is protected under strict data governance and privacy regulations (Conduah et al., 2025). Moreover, acquiring large-scale biomedical datasets such as imaging, fluid biomarkers, and genomics is expensive and requires coordination among multiple stakeholders, as well as rigorous study designs. These complexities not only limit the volume of data that can be collected but also affect its overall quality and consistency. This along with the difficulty of creating annotated datasets has lead to a shift towards techniques that bridge the gap between data scarcity and model development (Patki et al., 2016; Abdalla et al., 2025; Alzubaidi et al., 2023). **Synthetic data** has emerged as a promising approach in this context, providing a means to augment limited datasets, balance class distributions, and generate realistic samples while preserving patient privacy (Rajotte et al., 2022). By capturing the statistical properties of real biomedical data, synthetic datasets can support the development of more robust, generalizable models without direct reliance on sensitive patient records.

Various **generative modeling** techniques have been explored for synthetic data generation (Bauer et al., 2024), including Gaussian Mixture Models (Pezoulas et al., 2022), Kernel Density Estimation (Chintapalli et al., 2024), Generative Adversarial Networks (GANs) (Che et al., 2017), Variational Autoencoders (VAEs) (Jeong et al., 2025), among others. These approaches are typically applied directly to the source dataset, where the goal is to learn its underlying probability density. Once the density is estimated, synthetic samples can be generated by drawing from the learned distribution. However, applying these standard approaches to healthcare data is challenging. Healthcare data is extremely nuanced, exhibiting variability across participants due to factors such as demographics, genetics, lifestyle, and disease status. Independently modeling patient and control distributions may fail to preserve the true differences between the groups, since disease effects are often subtle and intermingled with the natural variability observed in healthy individuals (Marquand et al., 2016). This challenge is particularly evident in neurological disorders, which are characterized by heterogeneous disease effects. Conditions such as Alzheimer's disease (AD) and schizophrenia show substantial **heterogeneity** in both clinical presentation and disease progression (Lam et al., 2013; Picardi et al., 2012). If **paired** observations were available, that is, counterfactual representations of the same participant with and without disease, studying heterogeneous disease effects would be straightforward. However, such paired data are inherently unavailable, as participants are either have the disease or do not. As such, attempts to explain and study this heterogeneity have increasingly focused on methods that explicitly model how the patient distribution deviates from the healthy control distribution, while disentangling the disease effects from confounding healthy variability (Dong et al., 2015; Chand et al., 2020; Yang et al., 2021). These approaches underscore a key principle for generative modeling of healthcare data: to produce synthetic datasets of healthy and diseased participants that are meaningful, it is crucial to explicitly separate disease-related variability from normal inter-individual variability, ensuring that the synthetic data accurately captures the subtle but systematic differences between the two datasets.

Uncovering how two datasets differ is a key focus of **contrastive analysis (CA)**, a subfield of representation learning (Louiset et al., 2024). CA aims to learn generative or latent factors that explain both the variation shared across a background dataset (e.g., healthy participants) and a target dataset (e.g., diseased participants), as well as the factors that explain variation specific to the target dataset. Variation shared between the datasets is generally referred to as **shared** or **common** variation, while variation unique to the target dataset is referred to as **salient** variation. Separating the salient factors from common factors enables controlled synthesis of background and target samples while also facilitating the study of variation unique to the target distribution, such as disease-related heterogeneity (Aglinskas et al., 2022). Moreover, this framework naturally supports the generation of paired data, making it possible to synthesize a healthy sample alongside its diseased counterpart with personalized disease effects. Here, the counterfactual healthy sample is a data-driven approximation that reconstructs an individual using only the shared latent factors while removing salient disease-related attributes. Importantly, these counterfactuals should not be interpreted as causal in the sense of structural causal models(Pawlowski et al., 2020); rather, it is a generative, representation-level construct that isolates target-specific variation from common variation. This distinction is important, as the goal of the CA framework is to disentangle and model differences between datasets, not to infer causal effects.

The CA framework has most commonly been implemented using variational autoencoders (Abid & Zou, 2019; Severson et al., 2019), where it is commonly referred to as contrastive VAE. In this paper, we propose to adapt this framework for synthesizing neuroimaging-derived features, specifically regional brain volumes obtained from T1-weighted structural MRI, of healthy and diseased samples. We chose regional brain volumes, as they are widely used biomarkers to study changes in brain structure and atrophy due to neurological disorders, and have been used for diagnosis, prognosis, and subtype discovery(Eskildsen et al., 2015; Chand et al., 2020). Although we focus on regional brain volumes, the proposed framework is modality-agnostic: any structured neuroimaging-derived feature vector (e.g., cortical thickness, diffusion-derived connectometry, or functional connectivity) can be encoded into common and salient latent factors. We restrict our evaluation to anatomical ROI volumes because (i) they are the most consistently available measurements across large-scale multi-cohort datasets, (ii) site/scanner harmonization methods for volumes are well established(Pomponio et al., 2020), and (iii) disease effects are well characterized, enabling clear validation.

To better capture variability across individuals, we leverage data aggregated from multiple studies to build this framework. Our contributions are summarized below:

1 **Adaptation of the CA framework for neuroimaging-derived features**: We apply contrastive analysis using variational autoencoders for synthesizing regional brain volumes of both healthy and diseased participants.

2 **Paired synthetic data generation**: The framework supports synthesis of paired healthy-diseased samples, enabling personalized simulations of disease effects for downstream analysis. We formally show how CA optimizes an objective analogous to maximizing the joint likelihood of the healthy and disease distributions.

3 **Isolating disease-related variability in mild cognitive impairment (MCI) and Alzheimer's Disease (AD)**: We show that applying CA to MCI and AD participants captures disease-related salient variation.

4 **Multi-study integration**: Our approach leverages data aggregated across multiple cohorts, improving the diversity and utility of the synthesized samples.

5 **Facilitating access to derived features**: By synthesizing neuroimaging-derived features that are otherwise tedious to obtain due to preprocessing and segmentation requirements, our method provides a practical and clinically useful alternative data source for research applications.

## 2 BACKGROUND AND RELATED WORKS

This work relates to contrastive analysis, variational inference, synthetic data generation, and parsing disease heterogeneity.

### 2.1 CONTRASTIVE ANALYSIS

Formally, as described in (Abid & Zou, 2019; Severson et al., 2019), given i.i.d. samples $\{x\}_{i=1}^{N_x}$ from a target distribution and samples $\{y\}_{j=1}^{N_y}$ from a background distribution, contrastive analysis aims to learn latent variables $(z, s)$, where $z \in R^{D_z}$ captures the **shared variation** between the target and background datasets, and $s \in R^{D_s}$ captures the **salient variation** that is unique to the target dataset. Under the variational autoencoder setting, two probabilistic encoders $q_{\phi_z}(z|x)$ and $q_{\phi_s}(s|x)$ approximate the posteriors over the two sets of latent variables z and s, respectively. Then it is assumed that each sample $x_i$ or $y_j$ is drawn from a conditional distribution $p_\theta(.|z, s)$, parameterized by unknown decoder parameters $\theta$. Typically for the background samples, the salient latent variables are manually set to 0, but modifications exist (Weinberger et al., 2022; Louiset et al., 2023). To optimize the framework, a variational lower bound is defined for the log-likelihoods of individual data points, such that:

$$\log p(x_i) \geq \mathbb{E}_{q_{\phi_z}(z), q_{\phi_s}(s)} \log p_\theta(x_i|z, s) - KL(q_{\phi_z}(z|x_i)||p(z)) - KL(q_{\phi_s}(s|x_i)||p(s)) \quad (1)$$

$$\log p(y_j) \geq \mathbb{E}_{q_{\phi_z}(z)} \log p_\theta(y_j|z, 0) - KL(q_{\phi_z}(z|y_j)||p(z)) \quad (2)$$

Here, KL is the Kullback–Leibler divergence and $p(z)$ and $p(s)$ are the prior distributions over the two sets of latent variables $z$ and $s$, respectively. With $p(z)$ and $p(s)$ assumed to be multivariate isotropic Gaussians $\mathcal{N}(0, I)$. The two encoders along with the decoder are trained by maximize the sum of the objective functions (1) and (2).

### 2.2 PAIRED DATA SYNTHESIS: SHARED OBJECTIVE WITH CONTRASTIVE VAE

To synthesize paired samples $(x, y)$, the joint distribution $p(x, y)$ must first be defined. However, since in practice, paired data is often unavailable or difficult to acquire, we need to learn the joint distribution using unpaired observations from the marginal distributions i.e. $x \sim p(x)$ and $y \sim p(y)$. Directly learning the joint distribution $p(x, y)$ from unpaired samples $\{x\}_{i=1}^{N_x}$ and $\{y\}_{j=1}^{N_y}$ is non-trivial since $x$ and $y$ are not independent. But variational inference can impose conditional independence using latent variables. We again assume that $z$ captures variation common to both $(x, y)$, while $s$ captures target-specific variation. For neuroimaging features, $z$ may encode demographic or scanner variability shared between healthy and diseased participants, while $s$ encodes disease-related variability (e.g., atrophy patterns in MCI or AD). Under these assumptions, the conditional joint distribution can be written as

$$p(x, y|z, s) = p(x|z, s)p(y|z) \quad (3)$$

Similarly, the approximate posterior distribution factorizes as

$$q(z, s|x, y) = q(z|x, y)q(s|x) \tag{4}$$

Using variational inference, the log-likelihood of paired samples can be written as

$$\log p(x, y) = \mathbb{E}_{q(z,s|x,y)} \log \frac{p(x, y, z, s)}{q(z, s|x, y)} + KL(q(z, s|x, y)||p(z, s|x, y))$$

$$\geq \mathbb{E}_{q(z,s|x,y)} \log \frac{p(x, y|z, s)p(z, s)}{q(z, s|x, y)} \text{ (since KL divergence is non-negative)}$$

$$\geq \mathbb{E}_{q(z|x,y)q(s|x)} \log \frac{p(x|z, s)p(y|z)p(z)p(s)}{q(z|x, y)q(s/x)} \text{ (using (3), (4))}$$

$$\geq \mathbb{E}_{q(z|x,y)q(s|x)} \log p(x|z, s) + \mathbb{E}_{q(z|x,y)} \log p(y|z) - KL(q(z|x, y)||p(z)) - KL(q(s|x)||p(s)) \tag{5}$$

This is a variational lower bound for log-likelihood of paired samples. This formulation shows that the joint log-likelihood decomposes into two reconstruction terms (for $x$ and $y$) regularized by KL penalties on the common and salient latents. Importantly, because the bound depends on $q(z|x, y)$, paired data is still required. However, if we explicitly decouple the posterior such that

$$q(z|x, y) = q(z|x) = q(z|y) \tag{6}$$

then the joint log-likelihood can be maximized using unpaired samples. This essentially means that the common latent can be inferred from either the background or target sample, such that it explains the variation common to both the background and target dataset. Equation (5) simplifies to:

$$\log p(x, y) \geq \mathbb{E}_{q(z|x,y)q(s|x)} \log p(x|z, s) + \mathbb{E}_{q(z|x,y)} \log p(y|z) - \frac{1}{2} KL(q(z|x)||p(z))$$
$$- \frac{1}{2} KL(q(z|y)||p(z)) - KL(q(s|x)||p(s)) \tag{7}$$

We can see that (7) is equivalent to the sum of (1) and (2), which is the objective of CA, albeit with a weaker penalty on the KL divergence terms. This shows that CA can be understood as implicitly maximizing a joint likelihood, even when trained only on unpaired data.

Recent CA methods with variational autoencoders already enforce this decoupling to some capacity. In (Weinberger et al., 2022), a maximum mean discrepancy (MMD) term is added to the overall objective so that the distribution of the common latent is same across target and background samples. Similarly in (Louiset et al., 2024), a mutual information term is added to the objective to maximize the mutual information between the common latent and the two datasets.

## 3 METHODS: CONTRASTIVE ANALYSIS FRAMEWORK FOR UNPAIRED-TO-PAIRED DATA SYNTHESIS

Building on the contrastive VAE framework detailed in Section 2.1, we develop a model for generating paired neuroimaging-derived features from unpaired healthy and disease samples. Our approach encodes each sample into a common latent $z$, capturing variation shared across datasets, and a salient latent $s$, capturing disease-specific variation. By explicitly decoupling these latent spaces, the model can leverage unpaired data while enabling the generation of realistic paired samples.

### 3.1 MODEL ARCHITECTURE

The general architecture of the model is illustrated in Figure 1. The model consists of two probabilistic encoders, $q_{\phi_z}$ and $q_{\phi_s}$, mapping input features into the common and salient latent spaces, respectively, and a shared decoder $p_\theta$ that reconstructs input features from the concatenated latent representations. For background (healthy) samples, the salient latent is fixed to zero, whereas both latents are inferred for target (disease) samples. The latent dimensionality and encoder/decoder architectures are chosen to balance reconstruction fidelity with latent disentanglement. More on this in the implementation details Section 4.2

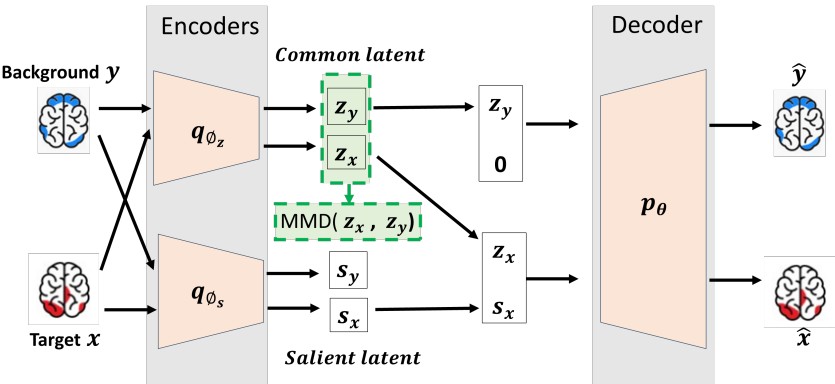

Figure 1: Illustration of the contrastive VAE model. The model separates latent variables into (i) common latents $z$, which capture variation shared between background (e.g., healthy) and target (e.g., diseased) data, and (ii) salient latents $s$, which capture variation unique to the target. Two encoders, $q_{\phi_z}$ and $q_{\phi_s}$, map inputs into these respective representations, which are concatenated and passed to a decoder $p_\theta$ for reconstruction. For background samples, the salient latents are fixed to zero. To enforce decoupling of the posterior distribution of the common latent, such that this latent captures shared variation across the target and background samples, maximum mean discrepancy between common latent representations of target ($z_x$) and background ($z_y$) is minimized during model training. Synthetic data are generated by sampling $z, s \sim \mathcal{N}(0, I)$ and decoding: paired target samples use both $z$ and $s$, paired background samples use $z$ with $s = 0$, and unpaired background samples use $z'$ sampled independently from $z$.

## 3.2 TRAINING OBJECTIVE

The model is trained to maximize a variational lower bound on the log-likelihood of target and background samples (see equation (1), (2). The training objective combines reconstruction terms, KL penalties on the latent posteriors, and a regularization term to align common latent distributions across datasets. This regularization term penalizes the MMD between $z_x$ and $z_y$ which are the common latent representation of the target and background samples, respectively. By doing so, we ensure that the common latent represents the variations shared between the target and background samples, ensuring that the common posterior under the paired setting is decoupled (see equation 6). So, the overall contrastive VAE objective that needs to be maximized is

$$\mathcal{L} = \sum_{i=1}^{N_x} \mathcal{L}_x(x_i) + \sum_{j=1}^{N_y} \mathcal{L}_y(y_j) - \lambda \, \mathrm{MMD}(z_x, z_y) \tag{8}$$

where $L_x(x_i) = \mathbb{E}_{q_{\phi_z}(z)q_{\phi_s}(s)} \log p_\theta(x_i|z, s) - KL(q_{\phi_z}(z|x_i)||p(z)) - KL(q_{\phi_s}(s|x_i)||p(s))$ is the variational lower bound for log-likelihood of target samples (see equation (1)), $L_y(y_i) = \mathbb{E}_{q_{\phi_z}(z)} \log p_\theta(y_j|z, 0) - KL(q_{\phi_z}(z|y_j)||p(z))$ is the variational lower bound for log-likelihood of background samples (see equation (2)), and $\lambda$ controls the strength of the alignment penalty. This objective encourages separation of common and salient latents while ensuring that the common latent captures variation shared across healthy and diseased participants.

## 3.3 SYNTHETIC PAIRED DATA GENERATION

Once the model is trained and the parameters $\phi_z$, $\phi_s$, and $\theta$ are learned, synthetic samples can be generated by sampling from the latent priors.

1) **Synthesizing paired background and target samples using latent priors**: Target samples are synthesized by decoding $(z, s)$, with $z \sim \mathcal{N}(0, I)$ and $s \sim \mathcal{N}(0, I)$. Paired background samples are synthesized by decoding $(z, s = 0)$.

2) **Conditional synthesis of paired background and target samples**:If unpaired real target and background samples are available, the paired counterpart can be synthesized conditionally. The common latent $z$ is inferred from the given sample, and to synthesize a paired target sample, $s \sim \mathcal{N}(0, I)$ is sampled and decoded with $z$: $p(x \mid z, s)$, and to synthesize a paired background sample, $s$ is set to 0 and decoded with $z$.

# 4 EXPERIMENTS

To implement the framework for synthesizing paired neuroimaging derived features from unpaired neuroimaging derived features, we first applied the framework to semi-synthetic data to validate its performance under controlled environment and then applied it to real data.

## 4.1 DATASETS

Both the semi-synthetic and real-data experiments used data from the iSTAGING consortium (Habes et al., 2021), which consolidated and harmonized imaging and clinical data from multiple studies spanning a wide age range (22–90 years). The consortium includes neuroimaging, demographic, and clinical measures from participants clinically diagnosed as cognitively normal (CN), with mild cognitive impairment (MCI), or with Alzheimer's disease (AD). The neuroimaging-derived features consist of 139 anatomical brain region-of-interest (ROI) volumes (119 gray matter ROIs and 20 white matter ROIs) extracted from baseline T1-weighted MRI scans using a multi-atlas label fusion method (Doshi et al., 2016). To harmonize data across studies and mitigate site effects, ComBat-GAM harmonization (Pomponio et al., 2020) was applied to the ROI volumes while adjusting for age, sex, and intracranial volume (ICV).

**Semi-Synthetic Experiment:** For this experiment, 3,900 cognitively normal participants (Age 62.9 $\pm$ 7.6 years, 52% female) were sampled from the iSTAGING dataset. Of these, 1,950 samples were used to construct a synthetic disease cohort with heterogeneous disease effects. Heterogeneity was simulated by defining three subtypes, each with 10–30% atrophy in predefined ROIs, with some overlap between subtypes (see Table 1).We applied 10–30% synthetic atrophy in the semi-synthetic data to reflect realistic disease effects observed in neurodegenerative conditions, ensuring clinically plausible variation while providing a meaningful challenge for generative modeling(Whitwell, 2010). The procedure was as follows: first, the 139 ROI volumes were covariate-corrected for age, sex, and ICV (volumes were residualized using linear regression). Then, for each synthetic disease sample $k$, one of the three subtypes was randomly assigned. For each ROI $l$ belonging to the chosen subtype, the corresponding volume $v_{kl}$ was reduced according to

$$v_{kl} = v_{kl} - \text{Uniform}(0.1, 0.3) \times v_{kl}.$$

Finally, the covariate effects were added back to preserve population-level structure. The cognitively normal and synthetic disease participants were respectively split into train (N=1050), cross-validation (N=450), and test (N=450) cohort for training and validating the contrastive VAE model. All volumes were standardized with respect to the cognitively normal training set.

**Real Data Experiment:** For the real-data experiment, we sampled 5,212 cognitively normal participants (Age 64.8 $\pm$ 10.5 years, 56.8% female) and 1,409 participants clinically diagnosed with MCI or AD (Age 77.0 $\pm$ 9.1 years, 53.7% female) from the iSTAGING dataset (independent from the semi-synthetic experiment). The healthy and diseased dataset were respectively split into training, cross-validation, and test cohorts using a 70:15:15 ratio. Because the healthy and disease cohorts have different demographic distributions, ROI volumes were covariate-corrected for age, sex, and ICV and the residuals were used in training the model. Specifically, volumes were residualized using linear regression parameters estimated from the cognitively normal training cohort. This procedure prevents the model from conflating demographic differences with disease-related effects. All ROI volumes were standardized with respect to the cognitively normal training set.

To further assess whether the trained model generalizes and captures disease-related salient variation, we evaluated it on an out-of-distribution (OOD) dataset from a study that was held-out from training. For this, we used the Alzheimer's Disease Neuroimaging Initiative (ADNI) dataset (Toga & Crawford, 2015) which has comprehensive clinical and cognitive measures that quantify disease burden. We selected 2,437 participants across ADNI1/2/GO/3 with baseline regional brain volumes.

Table 1: Bilateral ROIs included in disease patterns (OP:opercular part) in Semi-synthetic experiment, 10-30% atrophy simulated

| ROI | Atrophy Patterns | | |
|---|---|---|---|
| | S1 | S2 | S3 |
| Amygdala | ✓ | | |
| Temporal pole | ✓ | ✓ | ✓ |
| Hippocampus | ✓ | ✓ | |
| Angular gyrus | | | ✓ |
| Entorhinal area | ✓ | | |
| Frontal operculum | | ✓ | |
| Inferior temporal gyrus | ✓ | | |
| Lateral orbital gyrus | | | ✓ |
| Medial frontal cortex | | ✓ | ✓ |
| Middle frontal gyrus | | ✓ | |
| Middle occipital gyrus | | | ✓ |
| OP of the inferior frontal gyrus | | ✓ | |
| Parahippocampal gyrus | ✓ | | |
| Posterior insula | | ✓ | |
| Parietal operculum | ✓ | | ✓ |
| Supramarginal gyrus | | | ✓ |

This dataset comprised of 871 cognitively normal (Age 72.6 ± 6.3 years, 41.8 % female), 1,101 MCI (Age 72.8 $pm$ 7.6 years, 56.6% female), and 420 AD (Age 74.9 ± 7.8 years, 44% female) samples.

## 4.2 IMPLEMENTATION DETAILS

Since the inputs are one-dimensional regional brain volume features, both the common and salient encoders as well as the decoder were implemented as shallow networks consisting of two linear layers with LeakyReLU activations. Empirically we found that a latent dimensionality of 5 for both the common and salient variables provided sufficient capacity to capture the variability across the target and background datasets. The regularization coefficient $\lambda$ was set to 10 for the semi-synthetic experiments and to 100 for the real-data experiments. Max epochs is 500. For optimization, we used ADAM optimizer with learning rate 0.001. $\beta_1$ and $\beta_2$ are 0.5 and 0.999, respectively. To monitor the quality of synthesized healthy and diseased data during training, in addition to the reconstruction loss, we monitor the Wasserstein distance between the synthesized and cross-validation data distributions. Specifically, we use the 2-Wasserstein distance formula for two multivariate gaussian measures(Mallasto et al., 2022) to assess whether the generated data distribution is close to the observed ground truth reference distribution. We trained the model until the maximum number of epochs, unless a worsening trend was observed in the Wasserstein distance. Because the decoder outputs continuous ROI volumes, we model $p_\theta(.|z, s)$ as a multivariate Gaussian distribution with fixed identity covariance. Under this assumption, the decoder predicts only the mean of the distribution, and the reconstruction terms in the objective reduce to the mean squared error (MSE) between the input and the reconstructed output. This is a standard choice for continuous features, and it enables the model to focus on capturing the mean structure of the data while ensuring stable and well-behaved training.

## 5 RESULTS

We first evaluated the contrastive VAE model on the semi-synthetic dataset to assess its ability to capture disease-specific effects. As shown in Figure 2**a**, the model generates paired data that reflect the simulated disease patterns. Specifically, bilateral hippocampal atrophy was introduced in subtypes S1 and S2, and the generated paired samples successfully replicate these effects. The latent representations further support these findings. Figure 2**b** shows that the common latent dimensions are well-aligned between healthy and disease samples, while the salient latent dimensions encode subtype-specific information. In particular, the first four salient dimensions provide a structured representation of subtypes S1–S3.

To assess the contribution of the contrastive structure and the MMD alignment term, we performed an ablation study (see Appendix A.1 for full details). Table A1 shows that the CA-VAE with MMD regularization generates synthetic target and background samples with higher fidelity compared to

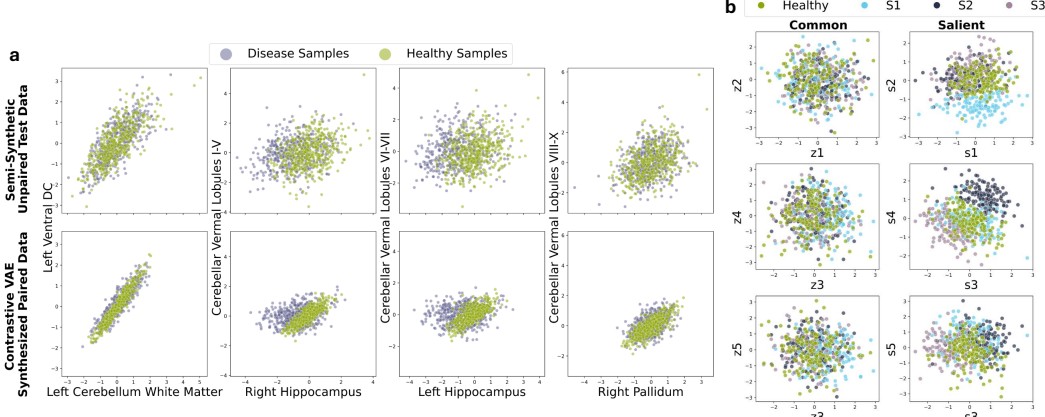

Figure 2: Semi-Synthetic experiment results: (a) Distribution of standardized brain volumes in the semi-synthetic test dataset (top row) and the contrastive VAE–generated paired dataset (bottom row). The bilateral hippocampus is included in subtypes S1 and S2. (b) Scatterplot of common and salient latent means when the semi-synthetic test dataset is input to the encoders. {z1,..,z5} and {s1,..,s5} denote the five common and salient dimensions, respectively.

a standard conditional VAE and to variants without MMD. Increasing the MMD penalty improves latent-space decoupling, as measured by a logistic regression classifier on the common latent means (AUC closer to 0.5 indicates better decoupling, Figure A1a). Additionally, K-means clustering on the salient latents demonstrates that the CA-VAE with MMD $\lambda = 10$ captures disease-subtype heterogeneity more effectively than the baseline model (Figure A1b). These results confirm that both the contrastive structure and the MMD alignment term are important for generating decoupled and subtype-informative latent representations.

After validating the model in a controlled setting, we trained it on real data. Figure 3**a** shows that real data are noisy, and even after covariate correction, the distributions of healthy and disease samples exhibit substantial overlap. Despite this, the model learns to synthesize paired healthy and disease samples that highlight structural variations in the bilateral hippocampus and amygdala, regions well-documented in the AD literature (Qu et al., 2023). Additionally, the common latent representation remains well-aligned across samples, while salient dimension 4 captures disease-related variation (Figure 3**b**). In this clinical cohort, the common latent dimensions were generally well-aligned across healthy and disease samples; however, common dimension 4 showed some separation between CN and AD subjects, suggesting subtle differences captured in the shared latent space. Given that ADNI is a clinical cohort, some confounding differences between healthy and disease participants are possible. As expected, salient dimension 4 captured variation associated with MCI and AD. Overall, the salient latent space captures disease severity: CN and MCI samples show considerable overlap, whereas AD samples are more distinct (Figure 4**a**), consistent with MCI representing early cognitive symptoms that may progress to AD.

We further leveraged the model's conditional synthesis capability to generate counterfactual healthy samples for each individual in the ADNI dataset. By computing the mean absolute difference (MAD) between observed and synthesized volumes, we observed a clear trend corresponding to disease severity: MAD was lowest for CN, higher for MCI, and highest for AD (Figure 4**b**). This demonstrates that the model has learned to disentangle shared from disease-related variation and can provide quantifiable measures of disease severity at the individual level.

## 6 DISSCUSSION

In this paper, we adapted a contrastive VAE framework for generating synthetic neuroimaging-derived brain regional volumes, providing a generative model that can help overcome challenges of limited sample sizes, imbalanced datasets, and privacy constraints. By making this model available,

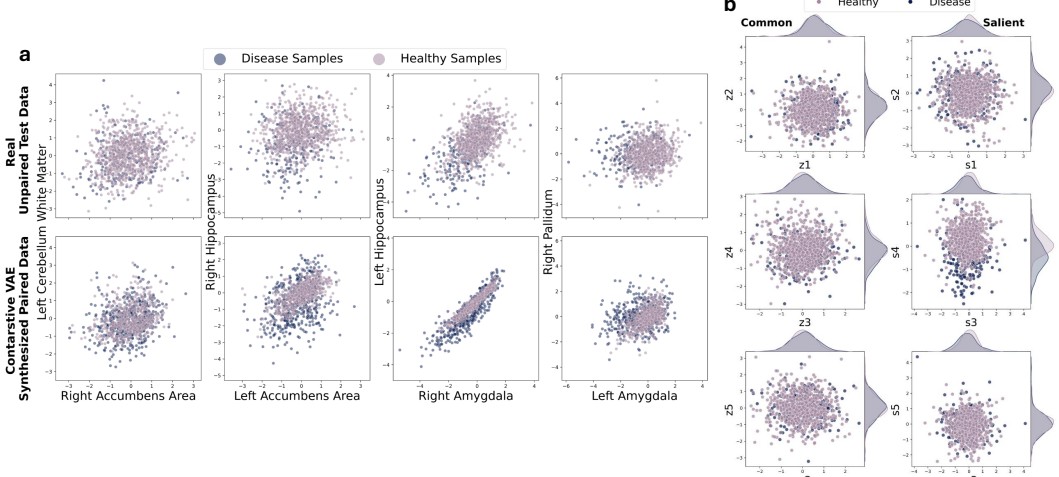

Figure 3: Real data experiment results: (a) Distribution of standardized brain volumes in the real test dataset (top row) and the contrastive VAE–generated paired dataset (bottom row). Atrophy can be seen in bilateral hippocampus and amygdala. (b) Scatterplot of common and salient latent means when the real test dataset is input to the encoders. $\{z1,..,z5\}$ and $\{s1,..,s5\}$ denote the five common and salient dimensions, respectively.

we aim to support research, education, and broader access to neuroimaging data, which often has a steep learning curve and significant preprocessing demands.

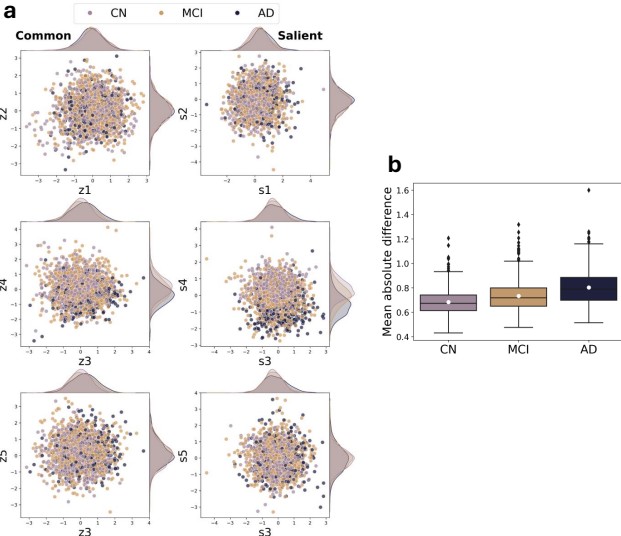

Figure 4: Results from OOD ADNI dataset: (a) Scatterplot of common and salient latent means when the ADNI dataset is input to the encoders. $\{z1,..,z5\}$ and $\{s1,..,s5\}$ denote the five common and salient dimensions, respectively. b) Boxplot of the mean absolute difference between observed brain volumes and contrastive VAE–synthesized healthy brain volumes. Healthy volumes were generated by combining each sample's inferred common latent representation with a salient latent set to zero, which is then passed through the decoder.

We showed that, under certain conditions, the objectives of contrastive analysis and paired data synthesis are equivalent, and used maximum mean discrepancy to achieve this alignment. By generating paired healthy and diseased samples, the model enables personalized modeling of disease effects, allowing fine-grained exploration of individual differences. This approach can be leveraged

for studying disease heterogeneity and subtype discovery. We demonstrated this capability on OOD ADNI data where salient latent dimensions captured variation associated with MCI and AD, reflecting disease-specific effects, while common dimensions preserved shared anatomical structure. Moreover, conditional synthesis of counterfactual healthy samples revealed an increasing trend in mean absolute differences between observed and counterfactual volumes across CN, MCI, and AD participants. This shows that the generalized well to OOD data and can be used to quantify disease severity at an individual-level.

Limitations of our current approach include its dependence on input data quality and the exclusive focus on regional brain volumes, which may not capture all aspects of disease effects. The model was evaluated primarily on a single dataset, and applying it to multiple cohorts could further test its generalizability. Additionally, incorporating regularization strategies based on known disease effects could further improve the fidelity of synthetic data. While the current latent representations are interpretable, they do not explicitly model discrete disease subtypes. Future work could extend the framework using deep clustering approaches combined with discrete salient representations, providing more control over subtype-conditioned synthetic data generation. This would allow more targeted simulations, enhance personalized modeling of disease effects, and provide a framework for exploring complex heterogeneity in neurological disorders. In summary, our framework provides a foundation for synthetic neuroimaging data generation that is practical, interpretable, and clinically relevant. By generating paired healthy and diseased samples, it enables personalized modeling of disease effects, supports subtype discovery, and facilitates the study of disease heterogeneity. With further extension to other multi-cohort datasets and target-informed regularization of the model, this approach has the potential to address challenges related to limited sample sizes, imbalanced datasets, and privacy concerns, while providing a valuable resource for research, education, and broader access to neuroimaging-derived data.

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

# A  APPENDIX

## A.1  ABLATION ANALYSIS OF THE CA FRAMEWORK AND MMD ALIGNMENT

To assess the utility of the contrastive autoencoder (CA) structure and the MMD alignment term, we performed an ablation study comparing several model variants: 1) Conditional VAE (cVAE)(Sohn et al., 2015) – a baseline without the contrastive structure, included to evaluate the added value of CA. 2) CA-VAE with $\lambda = 0$ – the CA structure without any MMD alignment, i.e., no decoupling of the shared latent space. 3)CA-VAE with $\lambda = 5$, $\lambda = 10$, and $\lambda = 15$ – the CA structure with increasing MMD regularization strength. 5) CA-VAE with Total Correlation (TC) regularizer – replacing MMD with the TC loss described in the original contrastive VAE paper (Abid & Zou, 2019). We evaluated fidelity of synthesized samples using 2-Wasserstein distances (lower is better for fidelity; for background-target separation, values closer to the real separation are better). Bootstrap resampling ($N = 1000$) was used to compute 95% confidence intervals. All models were trained on the semi-synthetic data. Table A1 shows that Contarstiev VAE framework with MMD alignment penalty helps generate synthetic background and target samples with higher fidelity.

To assess whether the shared latent representation was successfully decoupled (i.e. Equation 6) under different configurations of the CA framework, we evaluated how well a simple logistic regression classifier could distinguish target from background samples using only the common latent means produced by the encoder $q_{\phi_z}$. The latent means provide the deterministic component of the posterior and are standard for assessing latent-space structure. The classifier was trained using 5-fold cross-validation on latent codes extracted from the semi-synthetic dataset. An AUC near 0.5 indicates random classification, meaning that the latent space cannot separate the two datasets and

Table A1: Ablation study of MMD alignment in semi-synthetic experiment

| Model Variant | Generated vs Real Target | Generated vs Real Background | Unpaired Background vs Target |
|---|---|---|---|
| Semi-Synthetic data | - | - | 20.36 |
| Conditional VAE | 59.45 (58.62-62.45) | 61.7 (60.79-64.66) | 12.48 (11.9-14.94) |
| Contrastive VAE, $\lambda = 0$ | 62.18 (60.94-65.29) | 51.33 (50.73-55.05) | 14.99 (14.4-16.81) |
| Contrastive VAE, $\lambda = 5$ | 59.98 (59.28-62.82) | 53.19 (52.25-57.26) | 15.01 (14.28-17.08) |
| Contrastive VAE, $\lambda = 10$ | 60.47 (59.73-63.15) | 51.83 (51.42-55.06) | 15.16 (14.48-17.35) |
| Contrastive VAE, $\lambda = 15$ | 62.05 (60.86-64.91) | 54.21 (52.99-58.43) | 14.86 (14.15-16.91) |
| Contrastive VAE, TC loss | 61.14 (60.2-64.14) | 56.46 (55.34-60.9) | 13.58 (12.9-15.75) |

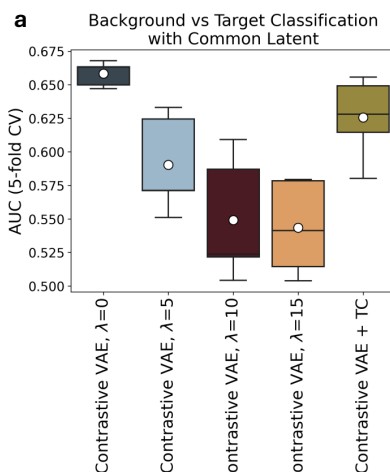 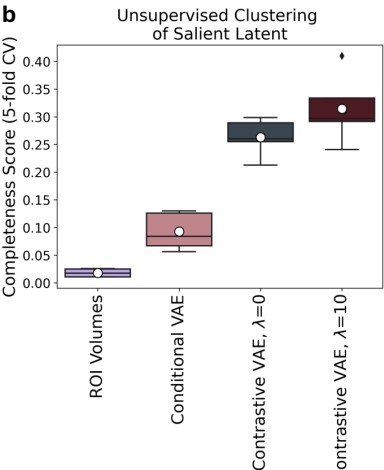

Figure A1: Supplementary results from semi-synthetic experiment: a) Boxplot of AUC scores from a logistic regression classifier trained to distinguish target from background samples using the common latent means. Lower AUC values indicate better latent-space decoupling across different Contrastive VAE configurations. b) Clustering performance of salient latent representations in the semi-synthetic dataset. K-means clustering was applied with k=4 (three disease subtypes plus background), and clustering accuracy was quantified using the completeness score across 5-fold cross-validation. Higher scores indicate better recovery of subtype-specific variation.

is therefore properly decoupled. Notably, when strength of alignment penalty $\lambda$ increases to 10 the mean AUC drops to 0.549, indicating improved decoupling (Figure A1a).

To quantitatively evaluate whether the latent space captures disease heterogeneity, we performed K-means clustering (with K=4, reflecting the three target subtypes plus the background in semi-synthetic data) on the salient latent means inferred from the semi-synthetic dataset. Clustering performance was assessed using the completeness score, with clustering repeated across 5-fold cross-validation to ensure robustness. The CA-VAE with MMD $\lambda = 10$ achieved the highest mean completeness score (0.315), outperforming both the raw ROI volumes and the latent space of the conditional VAE (Figure A1b). This indicates that contrastive analysis along with MMD alignment in the common latent space not only enforces decoupling but also enables the salient latent space to encode meaningful subtype-specific variation.

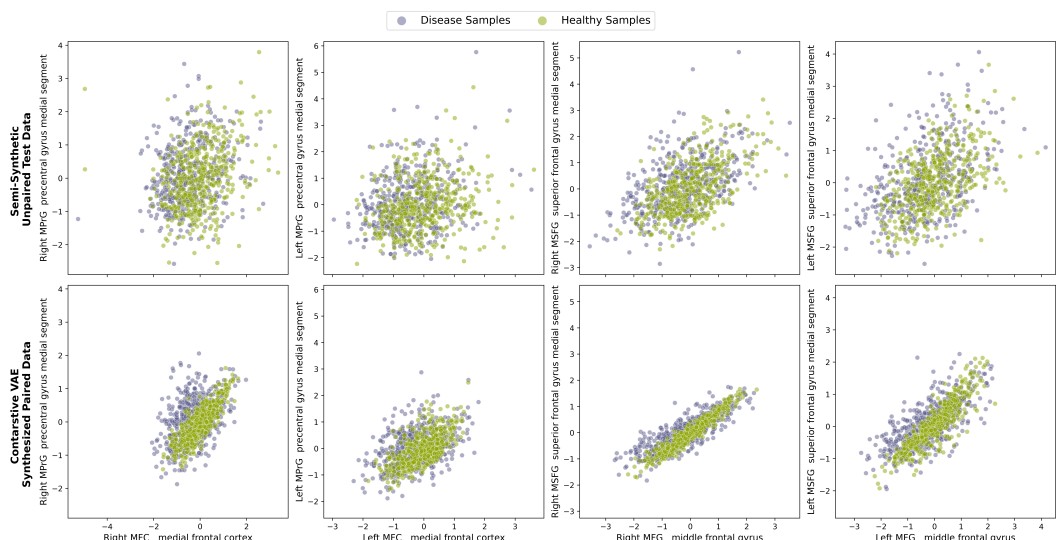

Figure A2: Supplementary results from semi-synthetic experiment: Distribution of standardized brain volumes in the semi-synthetic test data (top row) and the contrastive VAE–generated paired dataset (bottom row).

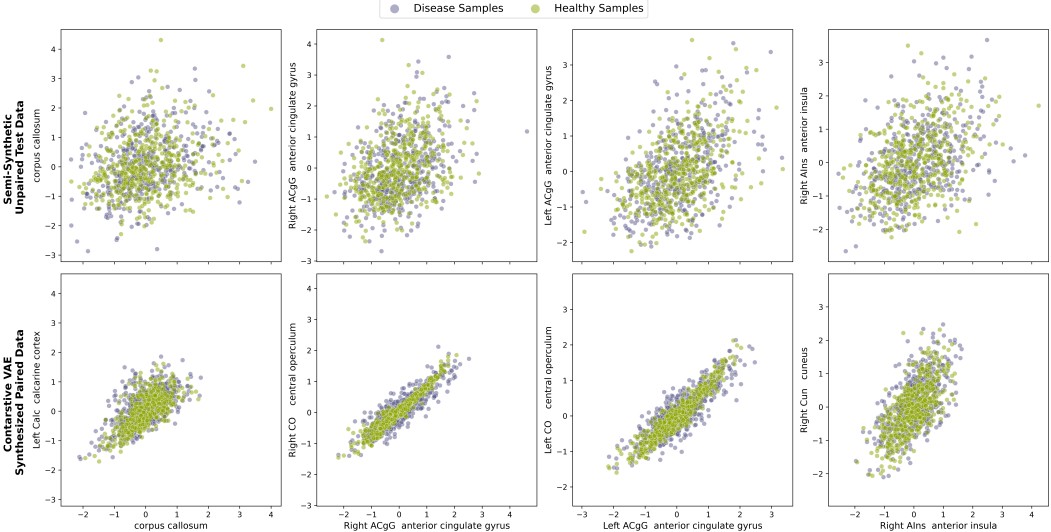

Figure A3: Supplementary results from semi-synthetic experiment: Distribution of standardized brain volumes in the semi-synthetic test data (top row) and the contrastive VAE–generated paired dataset (bottom row).

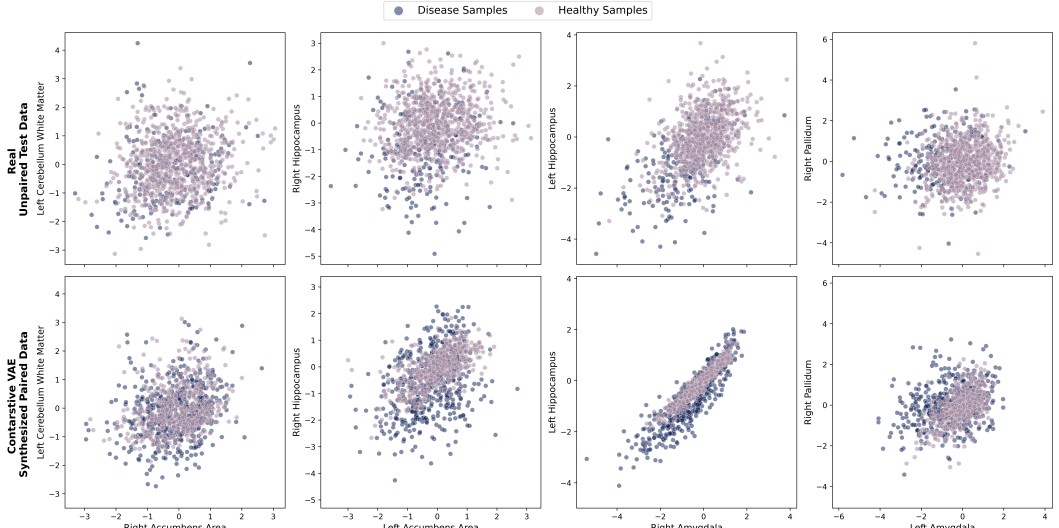

Figure A4: Supplementary results from real data experiment: Distribution of standardized brain volumes in the real test data (top row) and the contrastive VAE–generated paired dataset (bottom row).

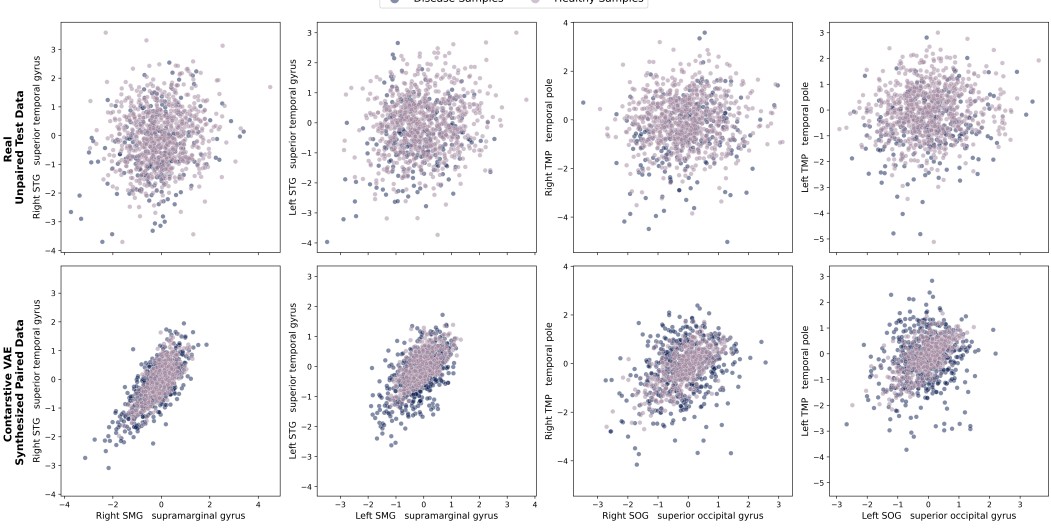

Figure A5: Supplementary results from real data experiment: Distribution of standardized brain volumes in the real test data (top row) and the contrastive VAE–generated paired dataset (bottom row).

