# OpenReview forum: "Unpaired-to-paired data synthesis: Learning to model disease effects via contrastive analysis of neuroimaging-derived features"
_ICLR.cc/2026/Conference — ICLR 2026 Conference Withdrawn Submission_

### Official Review · Reviewer_nRMY · 2025-10-30

**Soundness:** 1
**Presentation:** 2
**Contribution:** 1
**Rating:** 2
**Confidence:** 4

**Summary:**

This paper adapts the previously introduced contrastive VAE to generate realistic synthetic paired healthy and disease neuroimaging features from unpaired data. The method aims to disentangle “common” variation (shared between healthy and disease samples) from “salient” variation (disease specific effects) using two separate encoders and a shared decoder. The framework is trained on both real and synthetic (generated from healthy patients from the iSTAGING dataset) data (139 regional brain volumes) and tested on real (ADNI dataset) and synthetic cohorts. The authors generate paired synthetic data and qualitatively explore performance of their approach.

**Strengths:**

•	Using contrastive approaches to disentangle disease-specific from common variation in neuroimaging data is a reasonable approach to an important problem of generating realistic synthetic disease data


•	Using ComBat-GAM to adjust for site effects in a multi-site dataset is good practice.

**Weaknesses:**

Whilst the paper presents an interesting concept, I believe it is not suitable for publication in its current form due to the following major issues:

Major:

•	Novelty: The contribution is primarily an application of existing contrastive VAE methods to ROI volume synthesis, with no significant methodological advances.

•	Weak motivation for feature choice: The justification for using derived regional volumes rather than raw 3D images is unconvincing. Lines 111-114 claim the method "facilitates access to derived features that are otherwise tedious to obtain". However, image processing and segmentation pipelines such as Freesurfer and SynthSeg are readily available, automated and quick to run [5,6].

•	Methodology: The link between Equations 1 & 2 and Equation 5 is not explained. The paper claims Equation 5 is "equivalent to the sum of (1) and (2)" (line 181), but Equation 5 has one KL term for z while Equations 1+2 have two KL terms for z (one from x, one from y). How do these become equivalent under the decoupling assumption (Equation 6)? The connection between Equation 5 and the final objective (Equation 7) should be better justified. Also, the original contrastive VAE [4] includes a Total Correlation (TC) term (their Equation 5) to enforce independence between z and s latent variables. This paper does not appear to include this term, and it doesn’t appear that the authors have implemented an alternative.

•	Confusing logic:  It is unclear whether the model is trained on raw neuroimaging features or residuals (age, sex, ICV corrected). The semi-synthetic experiment adds covariates back after being regressed out (lines 286-291), but this is not mentioned for real data. If trained on raw features: It would make sense to match demographics between healthy/disease samples, but this doesn't appear to be done. If trained on residuals: What is z capturing? Can the model generate volumes for specific demographics? Otherwise the practical utility of this method seems limited.

•	No baseline evaluation: There are no baseline comparisons in the paper or quantitative analysis to evaluate how well their method is working. I appreciate there may be limited/no AD specific synthetic data generation methods existing in the literature, however there are several methods (eg. [7,8]) which generate healthy counterfactuals of disease images. The authors should compare their method to these baselines in the AD to healthy scenario for the ADNI dataset.

•	Questionable performance and limited results: Figures 2a and 3a show synthetic distributions (bottom rows) appear to have notably less variance than original data (top rows), particularly for healthy samples. This could be attributed to setting s=0 for healthy samples. It would have been nice for the authors to include some statistical tests to quantify these effects.

Minor:

•	Typos and missing links: Abstract states "The models are available at: [link]" but no actual link is provided. Also, there are several typos throughout the text.

•	Assumptions: Setting s=0 for all healthy individuals assumes no "healthy-specific" variation exists. Is this realistic? Do we believe all healthy variation is captured by the adjusted demographic features? This is debatable.

•	Missing references: The authors should reference [1] and [2] in relation to synthetic disease sample generation, and [3] in background contrastive VAE methods.


References:

[1] Unraveling Normal Anatomy via Fluid-Driven Anomaly Randomization, Peirong Liu, Ana Lawry Aguila, Juan E. Iglesias, CVPR 2025

[2] Image-Conditioned Diffusion Models for Medical Anomaly Detection, Matthew Baugh, Hadrien Reynaud, Sergio Naval Marimont, Sarah Cechnicka, Johanna P. Müller, Giacomo Tarroni & Bernhard Kainz, UNSURE MICCAI 2024

[3] Private-Shared Disentangled Multimodal VAE for Learning of Hybrid Latent Representations, Mihee Lee, Vladimir Pavlovic, CVPR workshop 2021

[4] Contrastive Variational Autoencoder Enhances Salient Features, Abubakar Abid, James Zou, arXiv, 2019

[5] Freesurfer, Bruce Fischl, NeuroImage, 2012

[6] SynthSeg: Segmentation of brain MRI scans of any contrast and resolution without retraining, Benjamin Billota, Douglas N. Greve, Oula Puonti, Axel Thielscher, Koen Van Leemput, Bruce Fischl, Adrian V. Dalcab, Juan Eugenio Iglesias, Medical Image Analysis, 2023

[7] Unsupervised 3D out-of-distribution detection with latent diffusion models, Mark S. Graham, Walter Hugo Lopez Pinaya, Paul Wright, Petru-Daniel Tudosiu, Yee H. Mah, James T. Teo, H. Rolf Jäger, David Werring, Parashkev Nachev, Sebastien Ourselin, M. Jorge Cardoso, MICCAI 2023

[8] Unsupervised brain imaging 3D anomaly detection and segmentation with transformers, Walter H.L. Pinaya, Petru-Daniel Tudosiu, Robert Gray, Geraint Rees, Parashkev Nachev, Sebastien Ourselin, M. Jorge Cardoso, Medical Image Analysis, 2022

**Questions:**

•	Could the authors please explain how their objective (Equation 7) links to the original contrastive VAE objective [4]?

•	How are you encouraging/enforcing independence between z and s?

•	How does Equation 5 become equivalent to the sum of Equations 1 and 2?

•	Are you training on residuals or raw ROIs?

---

### Official Review · Reviewer_1uvg · 2025-10-31

**Soundness:** 3
**Presentation:** 3
**Contribution:** 3
**Rating:** 4
**Confidence:** 4

**Summary:**

The authors proposes a contrastive VAE that learns to generate paired healthy–diseased neuroimaging feature vectors (regional brain volumes) from unpaired cohorts, enabling individualized modeling of disease effects and synthetic data generation for downstream analysis and privacy-sensitive contexts. Trained on multi-cohort ROI volumes, it decouples common variation from disease-specific variation to synthesize counterfactual “healthy” or “diseased” versions per subject and claims a joint-likelihood equivalence under a decoupled common latent assumption.

**Strengths:**

1. The reformulation of contrastive analysis for synthesizing paired healthy–disease feature vectors from unpaired cohorts is clearly motivated and well-aligned with neuroimaging constraints around privacy, heterogeneity, and limited labels.​

2. The explicit decoupling of shared and salient latents with an alignment penalty and the conditional “counterfactual healthy” synthesis demonstrate an interpretable mechanism for individual-level effect modeling, with a clean narrative from objective to qualitative outcomes.​

3. The semi-synthetic design articulates heterogeneity through subtype-specific atrophy patterns, and the OOD evaluation on ADNI provides a reasonable first step toward generalization beyond the training distribution.​

**Weaknesses:**

1. The central theoretical statement—that contrastive analysis implicitly maximizes a joint likelihood for paired samples—relies critically on the strong assumption $$q(z|x,y)=q(z|x)=q(z|y)$$, which is unlikely to hold without stringent identifiability conditions or auxiliary supervision, and no guarantees or diagnostics are provided to verify this assumption in practice.

2. The paper references maximum mean discrepancy to align common latents but does not show that MMD minimization suffices to enforce the posterior decoupling required for the joint-likelihood “equivalence,” nor does it quantify residual dependence or provide identifiability analysis.

3. Claims of multi-study integration and facilitating access to derived features are not backed by controlled comparisons showing concrete advantages over standard VAEs, cVAEs, or recent contrastive-disentanglement variants under matched capacity and training budgets.

4.  Evaluation is primarily qualitative (distribution overlays, scatter plots) with a single scalar trend (MAD between observed and counterfactual healthy) rather than rigorous quantitative utility metrics, such as: downstream diagnostic/prognostic tasks trained on synthetic vs. real data, subtype recovery scores in semi-synthetic settings, or calibration/fidelity measures beyond Gaussian 2-Wasserstein approximations.

5. There are no comparisons to strong baselines: standard VAE, $\beta$-VAE, cVAE conditioned on group, domain-adversarial models, MMDCVAE/SEP-VAE variants, or diffusion-based tabular generators, leaving it unclear whether the contrastive objective is necessary for the reported effects.

6. Decoder likelihoods, variance parameterization, and reconstruction loss instantiation are under-specified, despite modeling continuous ROI volumes where likelihood choice materially affects disentanglement and fidelity.

7. The training monitor uses the closed-form 2-Wasserstein distance between Gaussians to compare synthesized and real distributions, which is a restrictive approximation for multi-modal, non-Gaussian feature distributions and is not validated against more faithful distances.

**Questions:**

1. No ablations are provided for $\lambda$ (MMD strength), latent dimensionalities, decoder likelihoods, or alternative alignment strategies (e.g., MI terms), which is essential to support the mechanism and robustness claims. It would be good if the authors can add few experiments related to this

2. The semi-synthetic experiment injects subtype atrophy via fixed ROI sets but does not report quantitative subtype recoverability (e.g., clustering metrics on salient latents). Wouldn't it make it difficult to judge whether the model learns heterogeneity beyond the engineered signal?

3. Doesn't the observation that a “common” latent dimension partially separates CN and AD contradicts the intended decoupling and suggests leakage of salient information into the shared space?

4. As far as I understand, the “paired” generation notion is partly semantic—paired diseased samples are created by reusing $z$ with sampled $s\sim \mathcal{N}(0,I)$, which is not a true counterfactual for a specific subject unless $s$ is inferred from that subject. Can authors clarify this?

---

### Official Review · Reviewer_gAaK · 2025-11-02

**Soundness:** 2
**Presentation:** 2
**Contribution:** 1
**Rating:** 4
**Confidence:** 3

**Summary:**

The manuscript proposes contrastive VAEs to generate paired neuroimaging features from unpaired healthy and diseased samples. It separates latent factors into shared and disease-specific components and aligns shared representations across groups using an MMD loss. Trained on iSTAGING and tested on ADNI, the model reproduces known Alzheimer's atrophy patterns and generates individualized "healthy" counterfactuals whose differences from observed data reflect disease severity.

**Strengths:**

- The manuscript addresses an application in neuroimaging by generating paired healthy and diseased features from unpaired data.
- The experiments have been performed on iSTAGING and ADNI datasets

**Weaknesses:**

The methodological novelty is limited. The proposed framework is conceptually and structurally very close to existing Contrastive Analysis VAEs, notably SepVAE (Louiset et al., 2024), which also disentangles common and salient latent factors between background (healthy) and target (diseased) datasets. Both methods employ the same latent partition for shared variation and for disease-specific variation, and optimize standard ELBO objectives with regularizers. The main differences, replacing mutual information and classification regularizers (as in SepVAE) with an MMD alignment term, represent incremental adaptations rather than fundamentally new ideas.

While both methods are grounded in the same CA-VAE framework, only the present paper applies the model to Alzheimer's disease, whereas SepVAE focuses on psychiatric MRI and radiological datasets.  To strengthen the paper, the authors should include direct empirical comparisons with SepVAE and other recent CA-VAE variants to clarify the practical benefits of the proposed formulation.

Louiset, Robin, et al. "SepVAE: a contrastive VAE to separate pathological patterns from healthy ones." Medical Imaging with Deep Learning. PMLR, 2024.

**Questions:**

The paper describes the generated healthy versions as counterfactuals, but it is unclear whether these samples represent true causal counterfactuals or simply conditional reconstructions obtained by setting the disease-related latent variable to zero. In my understanding, since the model does not include an explicit causal structure or intervention mechanism, these generated samples should be interpreted as conditional generative reconstructions rather than causal counterfactuals. Could the authors clarify the assumptions or provide evidence that supports a causal interpretation, or acknowledge this limitation?

---

### Official Review · Reviewer_cpun · 2025-11-04

**Soundness:** 3
**Presentation:** 1
**Contribution:** 1
**Rating:** 4
**Confidence:** 4

**Summary:**

This paper revisits the Causal Autoencoder (CA) framework and reformulates it as a joint-likelihood maximization problem to enable paired synthesis of healthy–disease brain morphometric data, even when only unpaired samples are available. Using semi-synthetic atrophy simulations and real multi-cohort ROI-volume data (ADNI, etc.), the authors show that the model can disentangle “shared” vs. “salient” latent components, and that the magnitude of counterfactual differences (MAD) increases along disease stages (HC → MCI → AD). The study claims that such paired synthesis can support investigation of disease heterogeneity.

**Strengths:**

- Conceptually clean reinterpretation of the CA objective as a joint ELBO, connecting unpaired and paired formulations.
- Demonstrates the ability to synthesize subject-specific counterfactuals under an unpaired training regime.
- Consistent stage-dependent behavior of the salient component (MAD ↑ from HC to AD) suggests internal coherence of the model.
- Application to multi-study ROI-volume data is practical and shows reasonable reconstruction quality.

**Weaknesses:**

- The notion of *disease heterogeneity* is not convincingly demonstrated: the model distinguishes different disease stages, but does not reveal within-disease subtypes or inter-subject variability patterns.
- Biological interpretability of the salient latent dimension is weak. Showing that it correlates with hippocampal/temporal volumes is descriptive, not explanatory. No linkage to biomarkers or clinical scores (MMSE, CDR-SB, CSF, etc.) is provided.
- Semi-synthetic atrophy assumptions (10–30 %) are ad-hoc and lack empirical justification or sensitivity analysis.
- Writing clarity and notation consistency are problematic (e.g., conditional symbols, duplicated phrases, typos like “mean maximum discrepancy”).
- The framework is largely an incremental combination of known elements (cVAE + CA + MMD alignment) without strong downstream validation or comparison to recent disentanglement baselines.
- **No ablation analysis is provided**—it remains unclear how much each component (MMD regularization, shared–salient split, reconstruction terms) contributes to the observed effects. This makes the causal interpretation of model behavior questionable.
- Clinical utility remains unclear—regional volume reconstruction alone does not translate to actionable diagnostic or prognostic outcomes.

**Questions:**

- How does the proposed formulation handle within-disease heterogeneity beyond disease-stage progression?
- Are the salient latents correlated with any biological or clinical measures beyond group-level differences?
- What empirical basis motivated the 10–30 % atrophy range in the semi-synthetic experiment?
- How sensitive are results to the strength of the MMD regularization or to the latent dimensionality?
- Could the model’s synthetic data improve downstream tasks (diagnosis, progression prediction) compared with real data alone?
- Did you perform any ablation to isolate the contribution of each loss term (e.g., removing MMD alignment or salient latent disentanglement)? If not, how can we be confident that the claimed effects truly arise from your proposed formulation rather than generic cVAE behavior?

---

### Official Review · Reviewer_2wWR · 2025-11-11

**Soundness:** 3
**Presentation:** 2
**Contribution:** 2
**Rating:** 4
**Confidence:** 3

**Summary:**

This paper proposes a contrastive variational autoencoder framework to generate paired healthy-diseased neuroimaging-derived features (regional brain volumes) from unpaired datasets. The approach adapts the contrastive analysis (CA) framework to synthesize counterfactual pairs, enabling the modeling of disease effects and heterogeneity without requiring paired data. The method is validated on both semi-synthetic and real-world structural MRI datasets. Results show that the model captures disease-related salient variations and can generalize across cohorts.

**Strengths:**

1. This paper proposes a solution for an important practical challenge -- limited access to paired or balanced patient-control data due to privacy and acquisition barriers. Synthetic paired data could facilitate downstream tasks like disease severity modeling or heterogeneity analysis.
2. The latent structure is clear and intuitive for neuroimaging applications, promoting interpretability and personalized modeling of disease effects. The MMD-based alignment for shared latents is well-motivated and aligns with prior literature.

**Weaknesses:**

1. The model is evaluated exclusively on sMRI-derived regional volume features. While this is a reasonable starting point, it limits the demonstrated generality of the proposed approach. Evaluations on additional modalities would strengthen the paper's claim of broad applicability in neuroimaging synthesis.
2. The experimental section relies heavily on qualitative visualizations and distributional plots. The paper would benefit from more quantitative analyses to support its claims. Metrics such as reconstruction accuracy, sample realism (e.g., FID, MMD distances), or downstream performance (e.g., disease classification or subtype prediction using synthetic data) should be reported to objectively assess the fidelity and utility of the generated samples.
3. The proposed framework appears as a straightforward adaptation of an existing contrastive VAE formulation to the neuroimaging domain. While this transfer is valuable, the paper does not sufficiently justify why CA is a particularly suitable framework for modeling neuroimaging data. A more explicit discussion is needed on: a) what aspects of neuroimaging data (e.g., high heterogeneity, non-linear disease progression, site variability) make CA a good fit; b) what assumptions of CA may not hold in this context, and c) what specific modifications were made to adapt the model for neuroimaging and why those changes are reasonable. Such clarification would help position this work beyond a direct cross-domain application.
4. Table 1 appears poorly formatted or incomplete and is difficult to interpret. Please ensure that the table is properly formatted and legible.

**Questions:**

same as weaknesses

---

### Author Response · Authors · 2025-12-03

We sincerely thank the reviewers for their thoughtful and detailed comments. All feedback was extremely valuable and has helped us clarify key points, strengthen the manuscript, and better motivate our methodological choices. We have attached the revised manuscript (revisions highlighted in red) along with the anonymized code. We provide a consolidated response addressing the main points below:

1. Limitation to regional volumes:
Our focus on regional brain volumes is motivated by their widespread use as anatomical biomarkers, availability across multi-cohort studies, and established harmonization procedures. Importantly, the proposed framework is modality-agnostic: it can be extended to cortical thickness, diffusion MRI, or functional connectivity. We have clarified this in the manuscript and emphasized why regional volumes are the most appropriate first domain for rigorous evaluation.

2. Why Contrastive Analysis is suitable for neuroimaging:
Neuroimaging data exhibit substantial inter-individual variability due to demographics, scanner differences, and biological heterogeneity, while disease effects are subtle and nonlinear. Standard generative models often collapse disease-specific effects into general variability. Contrastive analysis explicitly separates shared from salient variation, making it well suited to neuroimaging. Without this factorization, cohort differences can leak into the latent code, preventing separation of shared and disease-specific factors, as confirmed by our ablation results (Figure A1b).

3. “Counterfactual” terminology:
We use “counterfactual” in the generative modeling sense: generating a subject conditioned on shared anatomical factors but with disease-specific variation removed. This is not a causal intervention under a structural causal model, and we do not claim detection of causal effects. The manuscript now clarifies that the model performs data-driven approximations of healthy anatomy in a controlled generative setting (Introduction, Lines 86–92).

4. Choice of 10–30% atrophy in semi-synthetic data:
The 10–30% range reflects realistic effect sizes reported in longitudinal AD studies, where annualized hippocampal or amygdala atrophy can reach 8% and cumulative brain changes up to 3%. This range ensures clinically plausible disease severity, a sufficiently challenging generative task, and the ability to simulate heterogeneous subtype trajectories without unrealistic deformations. This clarification has been added to Section 4.1 (Datasets).

5. Ablation and comparison to baselines:
We conducted a thorough ablation study (Appendix A.1, Table A1, Figure A1) to isolate the contributions of the contrastive structure and MMD alignment:
a) Fidelity: CA-VAE with MMD regularization produces synthetic target and background samples with higher fidelity compared to conditional VAE and CA-VAE without MMD.
b)Latent-space decoupling: Increasing the MMD penalty improves separation of shared vs. salient latents, with AUC of a logistic regression classifier approaching 0.5.
c) Subtype heterogeneity: K-means clustering on salient latents shows that CA-VAE with MMD best captures subtype-specific variation.
These results confirm that both the contrastive structure and MMD alignment are necessary for generating realistic, decoupled, and subtype-informative latent representations, and that the observed effects are not due to generic cVAE behavior.
6. Clarification on how does Equation 5 become equivalent to the sum of Equations 1 and 2: We added equation 7 in section 2.2 to make the connection between the two clear.
7.  Are we training on residuals or raw ROIs: In semi-synthetic experiment we train on raw ROI volumes after they have been standardized with respect to the background training dataset. In real data experiment, we train on residuals after covariate correction as clarified in line 316.
8. Clarification of Decoder Likelihood and Reconstruction Loss: We thank the reviewer for pointing out that the decoder likelihood parameterization required clearer specification. In our implementation, the decoder models continuous ROI volumes using a multivariate Gaussian likelihood with fixed identity covariance. Under this assumption, the decoder predicts the mean of the distribution, and the reconstruction term simplifies to mean squared error (MSE), which is equivalent to the negative log-likelihood up to a constant factor. This is a standard and widely adopted choice for continuous features, as it provides stable training while allowing the model to focus on accurately capturing the mean structure of the data. We have clarified this under section 4.2 Implementation details.

---

### Note · Authors · 2026-01-01

I have read and agree with the venue's withdrawal policy on behalf of myself and my co-authors.